# Nectin Cell Adhesion Molecule 4 (NECTIN4) Expression in Cutaneous Squamous Cell Carcinoma: A New Therapeutic Target?

**DOI:** 10.3390/biomedicines9040355

**Published:** 2021-03-30

**Authors:** Yuka Tanaka, Maho Murata, Yoshinao Oda, Masutaka Furue, Takamichi Ito

**Affiliations:** 1Department of Dermatology, Graduate School of Medical Sciences, Kyushu University, Fukuoka 812-8582, Japan; yukat53@med.kyushu-u.ac.jp (Y.T.); muratama@dermatol.med.kyushu-u.ac.jp (M.M.); furue@dermatol.med.kyushu-u.ac.jp (M.F.); 2Department of Anatomic Pathology, Graduate School of Medical Sciences, Kyushu University, Fukuoka 812-8582, Japan; oda@surgpath.med.kyushu-u.ac.jp; 3Research and Clinical Center for Yusho and Dioxin, Kyushu University Hospital, Fukuoka 812-8582, Japan

**Keywords:** NECTIN4, cutaneous squamous cell carcinoma, antibody-drug conjugate, targeted therapy

## Abstract

Cutaneous squamous cell carcinoma (cSCC) is the second most common skin cancer, and its incidence is rising because of the aging population. Nectin cell adhesion molecule 4 (NECTIN4) is involved in the progression of tumors and has attracted interest as a potential therapeutic target. However, little is known about the expression and significance of NECTIN4 in cSCC. The aim of this study was to determine the expression and function of NECTIN4 in cSCC. Immunohistological NECTIN4 expression was investigated in tissues from 34 cSCC patients. Using an A431 human SCC cell line, the role of NECTIN4 in the regulation of cell–cell attachment and migration and proliferation was assessed. NECTIN4 was expressed in most cSCC tissues and on the plasma membrane of A431 cells. Silencing of NECTIN4 prevented cell–cell attachment and induced the expression migration-related molecules, leading to an increase in cell migration. Knockdown of NECTIN4 downregulated extracellular signal-regulated kinase signaling, decreased cyclin D1 expression, and inhibited cell proliferation. These results show that NECTIN4 is expressed in cSCC and functions in the regulation of cell–cell interactions, as well as in the migration and proliferation of SCC cells. NECTIN4-targeted therapy may serve as a novel and promising treatment for cSCC.

## 1. Introduction

Cutaneous squamous cell carcinoma (cSCC) is the second most common skin cancer in humans. The estimated incidence of cSCC is 1 million cases per year, with an estimated 9000 deaths annually in the US [1,2,3]. The number of cases of cSCC has been increasing yearly, likely due to the aging population. Exposure to the sun or ultraviolet light, age, fair skin, and immunosuppression are known risk factors for cSCC [3,4,5,6]. Moreover, other environmental exposures, namely pollutants, ionizing radiations, and heavy metals, are also possible pathogenic factors for cSCC [7]. Although localized cSCC is curable by surgical excision and is associated with low mortality, 5–20% of patients will develop advanced cSCC, which has a worse prognosis; the 10-year survival rate is <10% for patients with distant metastases [8,9,10]. Several approaches have been used to treat advanced, metastatic, recurrent, or unresectable cSCCs, including radiation therapy and classical chemotherapy regimens, but the effects of these approaches are modest [11,12]. Targeted therapy is currently attracting considerable attention as a novel approach for the treatment of cSCC, with epidermal growth factor receptor (EGFR) being one of the targets. Clinical trials of EGFR inhibitors such as cetuximab, gefitinib, and erlotinib have been conducted and have demonstrated a certain level of response in advanced cSCC [13,14,15,16,17]. However, the efficacy of EGFR inhibitors remains modest and there is a possibility of the development of resistance to EGFR inhibition [18]. Thus, alternative treatments for cSCC are needed.

Nectin cell adhesion molecule 4 (NECTIN4; also known as poliovirus receptor-related protein 4 (PVRL4)) is a single-pass type I immunoglobulin-like membrane protein mainly found in adherens junctions [19]. NECTIN4 is a member of the nectin family, which regulates various cell functions, such as proliferation, differentiation, and migration [20,21]. NECTIN4 also plays a role in tumor progression by modifying cell proliferation, apoptosis, metastasis, and angiogenesis [22,23]. Because NECTIN4 expression is enhanced in several types of cancers, it is expected to be a novel target of cancer therapy [24,25]. Indeed, there are several ongoing clinical trials investigating the use of enfortumab vedotin, a NECTIN4-targeted antibody-drug conjugate (ADC), in urothelial cancer [26,27], as well as in locally advanced or metastatic malignant solid tumors (EV-202, NCT04225117) [28]. In normal skin, NECTIN4 is observed in the epidermis and skin appendages [29,30,31]. We recently reported that high NECTIN4 expression is associated with a poor prognosis of skin tumors such as extramammary Paget’s disease and malignant melanoma, and suggested the potential of NECTIN4 as a target in skin cancer therapy [31,32]. At present, the association between NECTIN4 and cSCC is not known; thus, the aim of the present study was to explore the expression and function of NECTIN4 in cSCC using clinical samples and human cSCC cells.

Here, we report that NECTIN4 is expressed in tissues from cSCC patients. In vitro assays showed that NECTIN4 regulates cell–cell interaction, as well as the migration and proliferation of cSCC cells. Our findings reveal the expression and function of NECTIN4 in cSCC and suggest the potential of NECTIN4 as a therapeutic target as a novel strategy for the treatment of cSCC.

## 2. Materials and Methods

### 2.1. Reagents

Monomethyl auristatin E (MMAE; ChemScene, Deerpark, NJ, USA; CS-0837), a widely used cytotoxic component of ADCs including a NECTIN4-targeted ADC enfortumab vedotin, and the extracellular signal-regulated kinase (ERK) inhibitor U0126 (Cell Signaling Technologies, Danvers, MA, USA; #9903) were dissolved in 0.1% dimethyl sulfoxide (DMSO; Sigma–Aldrich, St. Louis, MO, USA; 07-4860-5), which was also used as the vehicle control.

### 2.2. Cell Culture

A431 human squamous cell carcinoma (SCC) cells (ATCC, Manassas, VA, USA; ATCC^®^ CRL-1555™, RRID:CVCL 0037) were cultured in Dulbecco’s modified eagle’s medium (Sigma–Aldrich; D6429-500ML) supplemented with 5% fetal bovine serum (Nichirei Biosciences Inc., Tokyo, Japan; 174012) and penicillin–streptomycin (Thermo Fisher Scientific, Waltham, MA, USA; 15140-122). Normal human epidermal keratinocytes (NHEKs; Lonza, Basel, Switzerland; 00192907) were maintained in KGM™Gold™ Keratinocyte Growth Medium supplemented with KGM™Gold™SingleQuots™ supplements (Lonza; 00192060, 00192152). The medium was refreshed every 2 days. Cells were passaged at subconfluence and were used for experiments at fewer than five passages. Cell morphology was observed under a microscope and images were taken using a microscope camera system (Nikon Corporation, Tokyo, Japan; Nikon ECLIPSE TS100).

### 2.3. siRNA Transfection

To evaluate the function of NECTIN4 in SCC cells, A431 cells were seeded in 96-well plates (2000 cells per well), 12-well plates (1.2 × 10^5^ cells per well), or 6-well plates (2.0 × 10^5^ cells per well) and transfected with control siRNA (Invitrogen, Carlsbad, CA, USA; AM4611) or NECTIN4 siRNA (Invitrogen; s37689) using Lipofectamine RNAiMAX reagent (Invitrogen; 13778075) in accordance with the manufacturer’s instructions. Briefly, siRNAs were diluted in Opti-MEM™ I Reduced Serum Medium (Thermo Fisher Scientific; 31985062), mixed with Lipofectamine, and incubated for 20 min at room temperature. After the incubation, the siRNA–Lipofectamine complexes were mixed with the cells in culture plates (final siRNA concentration: 10 nM). The cells were harvested 1–5 days after transfection and used in further analyses. NECTIN4 knockdown efficiency was examined using quantitative reverse transcription–polymerase chain reaction (qRT-PCR) and western blotting.

### 2.4. Cell Proliferation Assay

Because the enfortumab vedotin, a NECTIN4-targeted ADC, is not commercially available at present, toxicity of its cytotoxic component MMAE on cell viability were assessed to estimate the efficacy of NECTIN4-targeted ADC. A431 cells or NHEKs were seeded in 96-well plates (2000 cells per well) and incubated for 24 h at 37 °C. The cells were then washed with Dulbecco’s phosphate-buffered saline (DPBS) before being further incubated in medium containing either 0.1% DMSO (vehicle control) or MMAE (1.25, 2.5, 5.0, 7.5, or 10 μM) for 48 h. To examine the effects of NECTIN4 knockdown on cell proliferation, cells were transfected with control or NECTIN4 siRNAs, as described above, seeded into 96-well plates, and incubated for 1–5 days. To examine the effects of ERK inhibition on cell proliferation, cells were seeded in 96-well plates and incubated with the ERK inhibitor U0126 (10 μM) for 1–5 days. After incubation, cells were treated with Cell Counting Kit-8 (CCK-8) solution (Dojindo, Kumamoto, Japan; 343-07623) for 2–4 h and absorbance was measured at 450 nm using a microplate reader (BioRad Laboratories, Hercules, CA, USA; 1681130J1).

### 2.5. Spheroid Formation Assay

To investigate the effect of NECTIN4 inhibition on cell–cell attachment, A431 cells were transfected with control or NECTIN4 siRNAs, as described above, seeded in 96-well clear, round-bottomed, ultra-low attachment microplates (Corning, Corning, NY, USA; #7007) at a density of 2000 cells per well, and incubated for 3 days at 37 °C. Spheroids were observed under a microscope and images were captured using a microscope camera system (Nikon Corporation). Projection area and the circularity of the spheroids were analyzed using ImageJ software (NIH, Bethesda, MD, USA).

### 2.6. RNA Extraction and qRT-PCR

Cells were collected and their RNA was extracted using an RNeasy Mini Kit (Qiagen, Hilden, Germany; 74104). The extracted RNA was converted to cDNA using a PrimeScript RT Reagent Kit (TaKaRa Bio Inc., Kusatsu, Japan; RR037). PCR was performed using TB Green Premix Ex Taq™ (TaKaRa Bio Inc.; RR420) and the following thermal program: 95 °C for 30 s, followed by 40 cycles of 95 °C for 5 s and 60 °C for 20 s. The sequences of primers are listed in Appendix A. Expression levels of target genes were normalized against the C_t_ value of β-actin (*ACTB*), and relative expression against the control condition was calculated using the comparative C_t_ method.

### 2.7. Western Blotting

Western blotting was performed as described previously [33]. The immunological bands were visualized with SuperSignal West Pico Chemiluminescence Substrate (Thermo Fisher Scientific; 34580) and detected using the ChemiDoc™ XRS Plus System (Bio-Rad Laboratories Inc. Hercules, CA, USA). Band intensities were measured using ImageJ software. The specifics of antibodies used are listed in Appendix A.

### 2.8. Immunocytochemistry

To investigate the expression and localization of NECTIN4, cells were seeded in 8-well μ-slide (ibidi GmbH, Gräfelfing, Germany; 80826) and incubated for 3 days at 37 °C. After washing with DPBS, cells were fixed with cold acetone for 10 min and then blocked with 5% bovine serum albumin (BSA) for 30 min at room temperature. The cells were then incubated overnight at 4 °C with the following primary antibodies: Rabbit anti-human NECTIN4 (1:200, Abcam; ab235897) and mouse anti-human E-cadherin (1:200, BD Biosciences, San Jose, CA, USA; 610181, RRID:AB_397580). To assess the proliferative status of the cells by analyzing Ki67, siRNA-transfected cells and DMSO- or U0126-treated cells were seeded in 8-well μ-slide and incubated for 4 days. The cells were then washed with DPBS, fixed with 4% paraformaldehyde (Fujifilm Wako Pure Chemicals Corporation, Osaka, Japan) for 15 min, treated with 0.1% Triton X-100 (Sigma–Aldrich) in DPBS for 15 min, and then blocked with 5% BSA for 30 min at room temperature. The cells were further incubated overnight at 4 °C with the primary antibody (rabbit anti-Ki67; 1:400; Cell Signaling Technologies; #9129; RRID:AB_2687446). After incubation with the primary antibody, cells were washed three times with DPBS and then incubated for 30 min at room temperature in the dark with the following secondary antibodies: AlexaFluor^®^-546-conjugated goat anti-rabbit IgG (1:400; Thermo Fisher Scientific; A11010; RRID:AB_2534077) and AlexaFluor^®^-488-conjugated goat anti-mouse IgG (1:400; Thermo Fisher Scientific; A11001; RRID:AB_2534069). The cells were then washed three times with DPBS, covered with mounting medium with 4′,6′-diamidino-2-phenylindole (DAPI; Vector Laboratories, Burlingame, CA, USA; H-1200), and observed under a fluorescence microscope (Thermo Fisher Scientific; EVOS-FL).

### 2.9. Scratch Assay

To investigate the effect of NECTIN4 inhibition on cell migration, A431 cells were seeded in a collagen-coated 96-well ImageLock tissue culture microplate (Essen Bioscience, Ann Arbor, MI, USA; 4379) at a density of 2.0 × 10^4^ cells per well, mixed with siRNA–Lipofectamine complexes, and incubated for 48 h. The cell monolayers were then scratched using a wound maker (Essen Bioscience) and the area of the wound in each well was automatically captured every 2 h using an IncuCyte live cell imaging system (Essen Bioscience). The wound area relative to that at 0 h was monitored and recorded using IncuCyte software (Essen Bioscience).

### 2.10. Patients

To assess the expression of NECTIN4 in cSCC patient tissues, a retrospective review of cSCC patients attending Kyushu University Hospital was performed. This study was approved by the Ethics Committee of Kyushu University Hospital (approval no. 30-363) on 27 November 2018, and written informed consent was obtained from patients before they were included in the study. This study was conducted in accordance with the guidelines of the Declaration of Helsinki. Tissues from 34 cSCC patients who were diagnosed and treated at the Department of Dermatology, Kyushu University, between 2014 and 2017 were used. Clinical information was collected from the patients’ files and used for analysis. At least three experienced dermatopathologists confirmed the diagnosis of cSCC.

### 2.11. Immunohistochemistry

Formalin-fixed, paraffin-embedded cSCC tissues were obtained from the archives of Kyushu University Hospital. Immunohistochemical staining was performed as described previously [34,35]. Briefly, tissue sections were treated with the primary antibody (rabbit anti-human NECTIN4; 1:150; Abcam; ab192033) for 30 min at room temperature. After incubation with secondary antibody (N-Histofine Simple Stain AP MULTI, Nichirei Biosciences; 414261) for 30 min at room temperature, sections were further treated with FastRed II (Nichirei Biosciences; 415261) and counterstained with hematoxylin.

### 2.12. Evaluation of NECTIN4 Staining

Immunohistochemical NECTIN4 expression levels were evaluated using a semiquantitative approach with H-scores [36]. The intensity of NECTIN4 staining was classified on a scale from 0 to 3+ following the previous publication [31,32] with slight modifications: 0, no staining; 1+, weakly positive; 2+, moderately positive; and 3+, strongly positive. Intensity level in the epidermis was set as 1+ and used as an internal control of staining. NECTIN4 H-scores were calculated by multiplying the percentage of positive cells (0–100%) by the staining intensity (0–3+), with final scores ranging between 0 and 300. Two dermatologists (M.M. and T.I.), who were blinded to the patients’ clinical information, independently assessed NECTIN4 expression. Images were captured using a microscope (Nikon Corporation; ECLIPSE 80i).

### 2.13. Statistical Analysis

Experiments were independently performed at least three times. Results are presented as the mean ± SD, and statistical analyses were performed using GraphPad Prism7 software (GraphPad Software, San Diego, CA, USA). Normality was analyzed using the Shapiro–Wilk test before following statistical analyses. The significance of differences between two groups was assessed using Student’s unpaired two-tailed *t*-test, and if the distribution of samples was not following normal distribution, the Mann–Whitney U test was used. The significance of differences between three or more groups was assessed using one-way analysis of variance (ANOVA) followed by multiple comparisons. Two-tailed *p* < 0.05 was considered statistically significant.

## 3. Results

### 3.1. NECTIN4 Expression in Tissues from cSCC Patients

NECTIN4 expression was assessed in tissues from 34 cSCC patients. The median of the patients’ age was 83.0 years (range 29–99 years) and 61.8% were male. As shown in the representative immunohistological images in Figure 1A, NECTIN4 was expressed in the epidermis and in cSCC tissues. The mean H-score for NECTIN4 expression in cSCC patients was 78.5 (range 3–244; Figure 1B). NECTIN4 was observed on the membrane and in the cytoplasm of cells comprising the tumors, and most of samples were positive for NECTIN4.

### 3.2. NECTIN4 Expression in A431 Cells and the Effects of MMAE on Cell Viability

Because NECTIN4 was expressed in tissues from cSCC patients, we assessed NECTIN4 expression and its function in vitro using the A431 human SCC cell line. A431 cells expressed NECTIN4 at both the mRNA and protein levels. NECTIN4 expression was significantly induced as the cells proliferated and became confluent (Figure 2A,B). Immunocytochemistry was used to examine NECTIN4 protein localization using simultaneous E-cadherin staining as a marker of cell membranes. Weak NECTIN4 staining was observed in the cytoplasm, whereas strong staining was observed on cell membranes, partly overlapping with E-cadherin (Figure 2C). We also investigated the effects of MMAE, a mitosis inhibitor comprising the cytotoxic part of NECTIN4-targeted ADCs, on the proliferation of A431 cells. The concentrations of MMAE used in this study were chosen on the basis of the MMAE concentration in patients’ peripheral blood reported previously [25]. After 48 h, the viability of MMAE-treated A431 cells was significantly decreased compared with that of DMSO-treated control cells, even at the lowest concentration of MMAE tested (1.25 nM), which is much lower than the concentration in patients’ peripheral blood (Figure 2D). Of note, NHEKs were less sensitive to MMAE than A431 cells. More specifically, viability of A431 cells treated with 1.25 nM of MMAE was significantly lower than that of NHEK cells (viability 29.3 ± 3.88% and 92.6 ± 7.92%, respectively, *p*-value less than 0.001).

### 3.3. Effects of NECTIN4 Silencing on Cell–Cell Interaction and Cell Migration

NECTIN4 is an adhesion molecule involved in cell-to-cell adhesion. Thus, we assessed the effect of NECTIN4 knockdown on cell adhesion. The efficiency of NECTIN4 knockdown was confirmed by qRT-PCR and western blotting; NECTIN4 expression at both the mRNA and protein levels was significantly inhibited on days 1–5 after siRNA transfection (Figure 3A,B). Over 60% of knockdown in mRNA level and over 55% knockdown in protein level was maintained during 5 days of analysis. Control siRNA-treated A431 cells grew tightly attached to each other under normal culture conditions. However, when NECTIN4 was inhibited, the adhesion between cells was weakened and the cells were separated from each other (Figure 3C). Tumor cells are able to grow in a spheroid shape, whereas normal cells cannot. In the spheroid form, tumor cells become less sensitive to anticancer drugs because the cells inside the spheroid are protected from the drugs. In this study, we used a spheroid assay with an ultra-low attachment culture plate to investigate the effects of NECTIN4 inhibition on spheroid formation. Cells in which NECTIN4 was silenced by siRNA transfection formed less compact spheroids with significantly larger projection areas than control cells (Figure 3D,E). In addition, cells in which NECTIN4 was silenced had a much more complex shape than control siRNA-transfected cells, as indicated by their significantly lower circularity (Figure 3E).

Because NECTIN4 knockdown attenuated cell–cell interactions, we investigated whether cell migration is increased by NECTIN4 silencing. There was a significant induction in the expression of zinc finger E-box binding homeobox 1 (ZEB1), zinc finger E-box binding homeobox 2 (ZEB2), and snail family transcriptional repressor 1 (SNAIL), all of which are involved in the regulation of cell migration, at both the mRNA and protein levels in NECTIN4-inhibited compared with control siRNA-transfected cells (Figure 4A,B). A cell migration assay was used to investigate whether the upregulation of these migration-related molecules led to increased cell migration. As shown in Figure 4C, cell migration was significantly increased in cells transfected with NECTIN4 siRNA compared with control siRNA. These results suggest that NECTIN4 has a role in the regulation of cell–cell adhesion, as well as in the migration of A431 cells.

### 3.4. Effects of NECTIN4 Silencing on the Proliferation of A431 Cells

NECTIN4 has been reported to promote cancer cell proliferation in tumors, including breast, gastric, and papillary thyroid cancers [37,38,39]. To further investigate the effects of NECTIN4 on cancer cell proliferation, we transfected cells with siRNA and then assessed cell proliferation. The number of viable cells was significantly decreased in NECTIN4 siRNA-transfected cells compared with control siRNA-transfected cells (Figure 5A). In addition, the expression of cyclin D1, a marker of cell proliferation, was significantly downregulated in NECTIN4 siRNA-transfected cells compared with control siRNA-transfected cells (Figure 5B,C). We also assessed the expression of Ki67, another marker of cell proliferation, using immunocytochemistry. The proportion of Ki67-positive proliferating cells was significantly decreased following knockdown of NECTIN4 (Figure 5D). These results suggest that NECTIN4 may play a role in regulating the proliferation of A431 cells.

### 3.5. NECTIN4 Regulation of Cell Proliferation through ERK Signaling

To obtain further insights into the mechanisms underlying the inhibition of cell proliferation in NECTIN4 siRNA-transfected cells, the phosphorylation status of ERK and Akt (a downstream molecule of phosphatidylinositol 3-kinase: PI3K signaling)—both known to regulate tumor cell proliferation [37,38,39,40,41,42]—was assessed. When NECTIN4 was silenced, there was a significant downregulation in ERK phosphorylation compared with control siRNA-transfected cells. Conversely, NECTIN4 inhibition had no effect on Akt phosphorylation (Figure 6A). To further investigate the involvement of ERK signaling, studies were performed using the ERK inhibitor U0126. ERK inhibition by U0126 was confirmed by western blotting (Figure 6B), and ERK inhibition significantly decreased the proliferation of A431 cells (Figure 6C). In addition, ERK inhibition significantly decreased cyclin D1 expression (Figure 6D,E) and the proportion of Ki67-positive proliferating cells (Figure 6F), as seen in NECTIN4 siRNA-transfected cells.

## 4. Discussion

NECTIN4 is known to be highly expressed in several tumors and enhances tumor progression [23,24]. However, the expression and function of NECTIN4 in cSCC has been unknown. Here we reported, for the first time, that NECTIN4 is expressed in tissues from cSCC patients. Silencing of NECTIN4 attenuated cell–cell adhesion and led to increased motility of A431 human SCC cells. The results also suggest that NECTIN4 may regulate the proliferation of SCC cells by modulating cyclin D1 expression partly through ERK signaling.

Because NECTIN4 is an adhesion molecule expressed mainly on the cell membrane, it was expected that NECTIN4 levels would affect cell–cell attachment. Indeed, changes in NECTIN4 levels reportedly alter the status of cell adhesion, leading to changes in the spheroid formation and growth of cancer cells. For example, in ovarian cancer cells, NECTIN4 was shown to alter cell adhesion and, although control cells formed small and compact spheroids, cells in which NECTIN4 was inhibited formed loosely aggregated spheroids [43]. NECTIN4 also supports the tumor progression of breast cancer cells by promoting anchorage-independent growth and cell proliferation through cell–cell interactions [44]. Similar to these reports, in the present study, NECTIN4 inhibition attenuated the cell–cell attachment of A431 cells in the 2D culture (Figure 3C) and loosened the aggregation of spheroids (Figure 3D,E), implying that NECTIN4 also regulates cell–cell interactions in cSCC and may contribute to the growth of cancer cells.

As shown in this study, the proliferation of A431 cells was significantly inhibited by NECTIN4 knockdown, accompanied by a decrease in cyclin D1 expression. Cyclin D1 is a member of the cyclin family and regulates cell cycle transition from the G_1_ to S phase. Cyclin D1 is frequently overexpressed and/or amplified in many types of cancers, including cSCC [40,45,46]. Although the relationship between NECTIN4 and cyclin D1 expression has not been investigated to date, cyclin D1 is known to be regulated by various mechanisms, including ERK signaling [40,41,42]. For example, cyclin D1 is upregulated via an ERK and nuclear factor (NF)-κB/cAMP response element (CRE)-mediated transcriptional mechanism in intestinal epithelial cells [47], and via a pERK/pc-Jun pathway in smooth muscle cells [48]. A relationship between cyclin D1 and ERK signaling has also been reported in SCC cells. In a study using A431 cells, combination treatment with an ornithine decarboxylase inhibitor and a cyclooxygenase inhibitor decreased pERK, which was associated with decreased cyclin D1 expression and decreased cell proliferation [49]. In another SCC cell line, SCL-1, all-trans retinoic acid treatment reduced the ERK-activated protein 1 pathway, accompanied by decreased cyclin D1 expression [50]. In the present study, ERK phosphorylation was decreased in A431 cells following NECTIN4 knockdown, along with significant inhibition of cyclin D1 expression and cell proliferation. When ERK was inhibited using the ERK inhibitor U0126, cyclin D1 expression and cell proliferation were also inhibited in A431 cells. Although further studies are needed to reveal the detailed mechanisms as to how NECTIN4 functions, we speculate that ERK signaling may be involved, at least in part. Although NECTIN4 is known to regulate the PI3K/Akt pathway in several cancers, such as breast, gastric, thyroid, and melanoma [32,37,38,39], this pathway was not affected by NECTIN4 inhibition under the present experimental conditions.

In the present study, NECTIN4 inhibition significantly induced the expression of ZEB1, ZEB2, and SNAIL, resulting in increased cell migration. To date, the relationships between NECTIN4 and these molecules have not been reported. Generally, these molecules are known as regulators of epithelial-to-mesenchymal transition (EMT), a process in which endothelial cells acquire mesenchymal characteristics accompanied by downregulation of E-cadherin and upregulation of vimentin [51,52]. However, even though ZEB1, ZEB2, and SNAIL expression was induced in NECTIN4-silenced A431 cells, we did not observe any obvious changes in E-cadherin and vimentin expression (Appendix A). Because these molecules function not only as triggers of EMT, but also as inducers of cell migration [53,54,55], NECTIN4 may alter the migratory status of A431 cells by regulating ZEB1, ZEB2, and SNAIL without affecting EMT. In addition, EMT is regulated through complex systems and molecules [51,52], and additional triggers may be required to accomplish complete EMT in A431 cells.

We here suggest NECTIN4 as a potential therapeutic target because NECTIN4 will serve as a landmark for the drug to recognize NECTIN4-expressing cSCC cells. Since NECTIN4 also contributes to the viability and proliferation of tumor cells, inhibition of NECTIN4 itself might be another way to utilize NECTIN4 as a therapeutic target. We found, however, the effect of NECTIN4 inhibition on cell proliferation was not so strong. Because the proliferation of cancer cells is regulated by multiple factors, NECTIN4 might be one of those factors, but not an only responsible factor. Thus, effects of NECTIN4 knockdown on cell proliferation might be moderate. In addition, NECTIN4 rather affects cell viability to a higher extent than cell proliferation as observed in Figure 5. In several types of cancer cells, overexpression of NECTIN4 increases the viable cells and inhibition of NECTIN4 decreases it [38,39]. Further, downregulation of NECTIN4 is also known to induce cell apoptosis and lead to the decreased viability [39]. Thus, NECTIN4 may potentially be involved in the apoptosis in SCC cells affecting viability. Since the function and related factors of NECTIN4 in cell proliferation and apoptosis is largely unknown at present, more detailed mechanisms regarding how NECTIN4 affect cell viability and proliferation in coordination with other factors need to be investigated after time to emphasize the potential of NECTIN4 as a promising therapeutic target.

Our findings that A431 cells express NECTIN4 and are highly sensitive to MMAE imply that these cells may be sensitive to NECTIN4-targeted ADCs such as enfortumab vedotin. Enfortumab vedotin is a novel NECTIN4-targeting drug that is comprised of an anti-NECTIN4 monoclonal antibody, linker, and the antimitotic drug MMAE (so-called vedotin) [24,25,26,27]. Some clinical trials have investigated the efficacy of enfortumab vedotin for the treatment of urothelial cancer [26,27] and several malignant solid tumors [28]. If enfortumab vedotin is to be considered for the treatment of cSCC, attention needs to be paid to its adverse effects on normal skin because normal keratinocytes express NECTIN4 [31]. Because enfortumab vedotin is not currently commercially available as a reagent, in the present study we tested the effects of MMAE on A431 cells and NHEKs to determine whether these cells have different sensitivities to MMAE. We found that NHEKs were less sensitive to MMAE than A431 cells: The viability of NHEKs was not affected by MMAE at concentrations up to 2.5 nM, whereas the viability of A431 cells was significantly decreased by MMAE at a concentration as low as 1.25 nM (~70% decrease in cell viability; Figure 2D). Because A431 cells are cancer cells and proliferate very rapidly, whereas NHEKs are relatively less proliferative, A431 cells may react with antimitotic drugs more readily and more strongly than NHEKs do. Thus, the adverse effects of enfortumab vedotin on normal skin may be controlled, in part, by careful selection of doses that are toxic to rapidly proliferating tumor cells, but not to normal keratinocytes.

## 5. Conclusions

In conclusion, we have shown that NECTIN4 is expressed in human cSCC tissues and is involved in the regulation of cell–cell interaction, as well as the migration and proliferation of SCC cells. We also suggest NECTIN4-targeted therapy as a novel and promising treatment for cSCC.

## Figures and Tables

**Figure 1 biomedicines-09-00355-f001:**
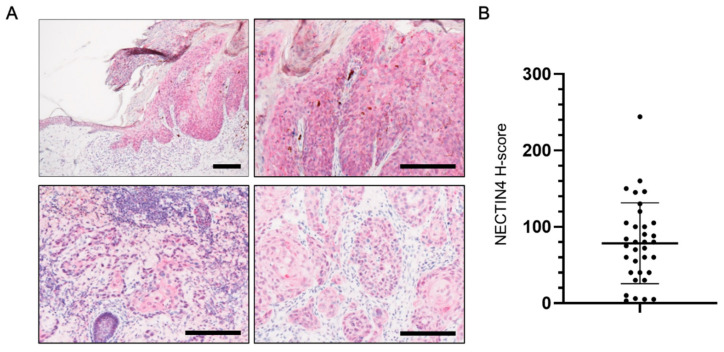
NECTIN4 expression in human cutaneous squamous cell carcinoma (cSCC) tissues. (**A**) Representative immunohistochemical images of NECTIN4 staining (red) in tissues from cSCC patients. (**B**) Mean (±SD) H-scores for NECTIN4 staining intensity in tissues from cSCC patients (*n* = 34). Symbols indicate H-scores in individual patients. Scale bars: 200 μm.

**Figure 2 biomedicines-09-00355-f002:**
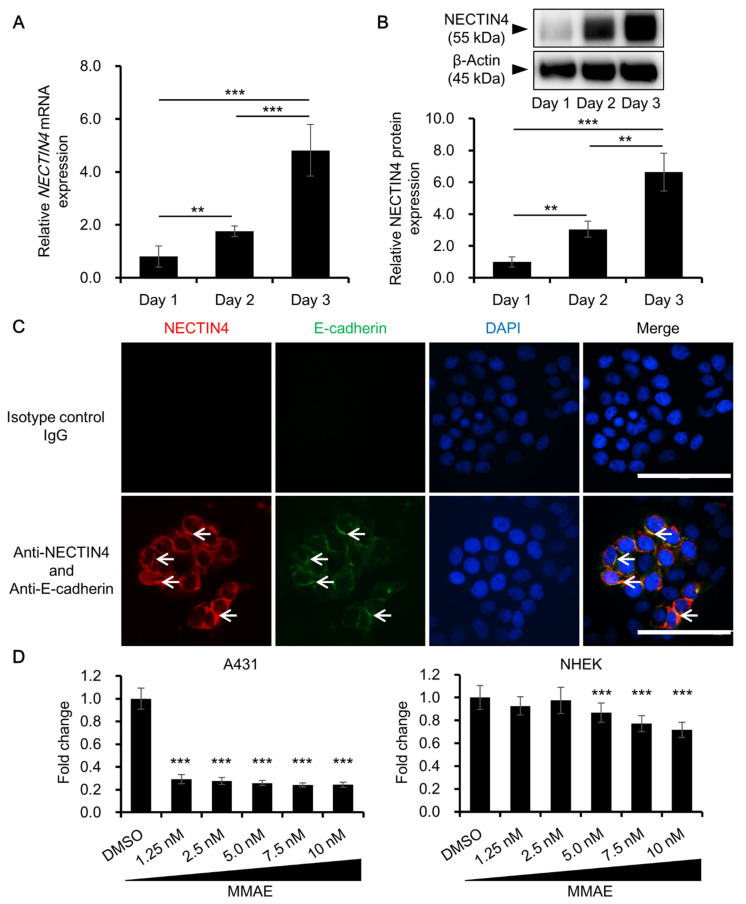
NECTIN4 expression in A431 human SCC cells. (**A**,**B**) Relative NECTIN4 expression in A431 cells at the mRNA (**A**) and protein (**B**) levels on days 1–3. Data are the mean ± SD (*n* = 3). ** *p* < 0.01, *** *p* < 0.001. (**C**) Representative immunocytochemical images of NECTIN4 and the membrane marker E-cadherin in A431 cells. Arrows indicate areas where NECTIN4 staining overlaps with E-cadherin staining. Scale bars = 100 μm. (**D**) Mean ± SD viability of A431 cells (left) and normal human epidermal keratinocytes (NHEKs) (right) treated for 48 h with dimethyl sulfoxide (DMSO) (0.1%) or different concentrations of monomethyl auristatin E (MMAE), as indicated. Experiments were independently repeated three times and performed in five wells for each concentration. *** *p* < 0.001 compared with DMSO (vehicle control).

**Figure 3 biomedicines-09-00355-f003:**
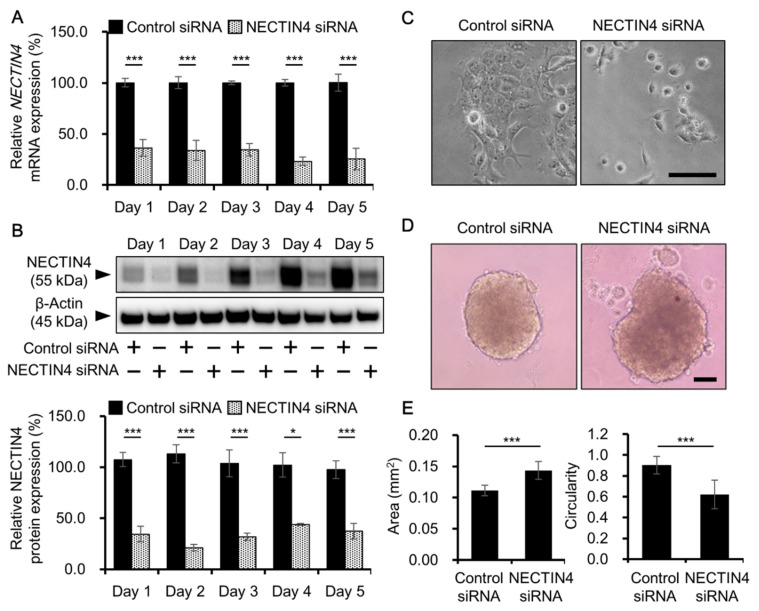
Effects of NECTIN4 inhibition on the adhesion of A431 cells. (**A**,**B**) NECTIN4 was inhibited in A431 cells by siRNA transfection. The efficiency of NECTIN4 knockdown at the mRNA (**A**) and protein (**B**) levels is shown on days 1–5 after siRNA transfection. Data are the mean ± SD (*n* = 3). (**C**) Representative microscopic images of control or NECTIN4 siRNA-transfected cells. Scale bars = 100 μm. (**D**) Representative images of spheroids derived from control or NECTIN4 siRNA-transfected A431 cells. (**E**) Area (left) and circularity (right) of spheroids derived from control or NECTIN4 siRNA-transfected A431 cells. Spheroid formation was independently repeated three times and evaluated in six wells per condition. Data are the mean ± SD. * *p* < 0.05, *** *p* < 0.001.

**Figure 4 biomedicines-09-00355-f004:**
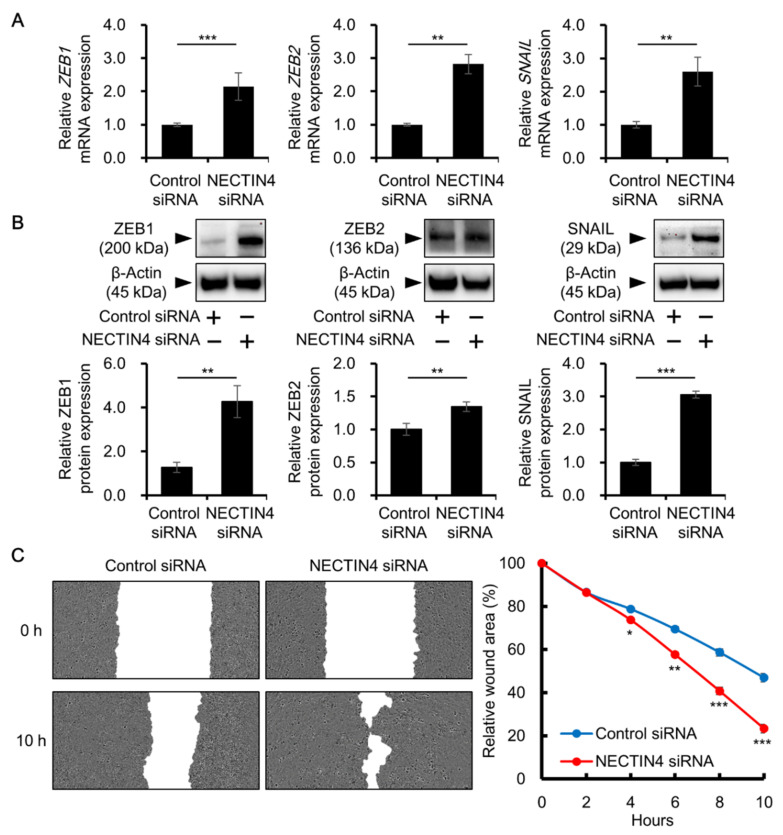
Effects of NECTIN4 inhibition on the migration of A431 cells. (**A**,**B**) Cells were transfected with control or NECTIN4 siRNA and the expression of ZEB1, ZEB2, and SNAIL was assessed at the mRNA level after 48 h (**A**) and at the protein level after 72 h (**B**). Data are the mean ± SD. (*n* = 3). ** *p* < 0.01, *** *p* < 0.001. (**C**) Monolayers of control or NECTIN4 siRNA-transfected A431 cells were scratched, and cell migration was monitored up to 10 h at 2-h intervals. Representative images of the wound area at 0 and 10 h are shown for control and NECTIN4 siRNA-transfected A431 cells. The graph shows the mean ± SD of relative wound area. The migration assay was independently repeated three times and performed in 18 wells per condition. * *p* < 0.05, ** *p* < 0.01, *** *p* < 0.001 compared with control siRNA-transfected cells.

**Figure 5 biomedicines-09-00355-f005:**
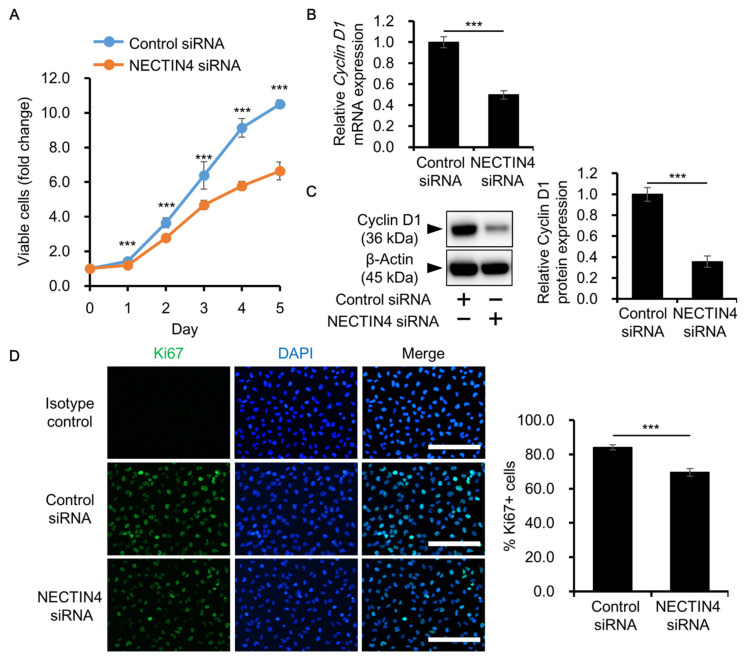
Effects of NECTIN4 inhibition on the proliferation of A431 cells. (**A**) Cells were transfected with control or NECTIN4 siRNA and cell viability was evaluated on days 1–5 after transfection using the CCK-8 cell viability assay. Experiments were independently repeated three times and performed in five wells per condition. *** *p* < 0.001. (**B**,**C**) Relative cyclin D1 expression at the mRNA (**B**) and protein (**C**) levels in control and NECTIN4 siRNA-transfected cells (*n* = 3). *** *p* < 0.001. (**D**) Representative immunocytochemical images of Ki67 staining (left) and the percentage of Ki67-positive cells (right) in control and NECTIN4 siRNA-transfected cells. DAPI was used to detect cell nuclei. Three different areas per well were randomly selected and three independent wells were observed. Scale bars = 200 μm. Data are presented as the mean ± SD. *** *p* < 0.001.

**Figure 6 biomedicines-09-00355-f006:**
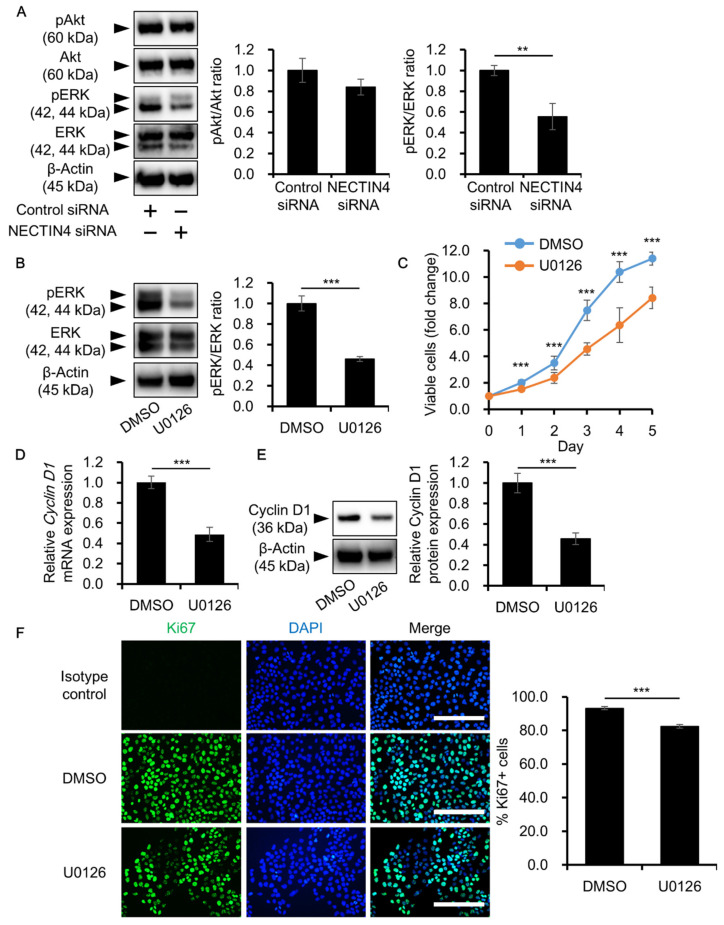
NECTIN4 regulation of A431 cell proliferation via ERK signaling. (**A**) The phosphorylation status of ERK and Akt was assessed in control and NECTIN4 siRNA-transfected cells. Representative blots are shown, in addition to the mean ± SD of pAkt/Akt and pERK/ERK ratios (*n* = 3). ** *p* < 0.01. (**B**) Inhibition of ERK phosphorylation by U0126 was confirmed by western blotting. Representative blots are shown, as well as the mean ± SD of pERK/ERK ratio in the DMSO (vehicle control)- and U0126-treated groups (*n* = 3). *** *p* < 0.001. (**C**) Viability of DMSO (0.1%)- and U0126 (10 μM)-treated cells on days 1–5 after treatment, as assessed using the CCK-8 cell viability assay. Experiments were independently repeated three times and performed in five wells per condition. *** *p* < 0.001. (**D**,**E**) Mean (±SD) relative cyclin D1 expression at the mRNA (**D**) and protein (**E**) levels in DMSO- and U0126-treated cells (*n* = 3). *** *p* < 0.001. (**F**) Representative immunocytochemical images of Ki67 staining (left) and the mean ± SD percentage of Ki67-positive cells (right) in DMSO- and U0126-treated cells. DAPI was used to detect cell nuclei. Three different areas per well were randomly selected and three independent wells were observed. *** *p* < 0.001. Scale bars = 200 μm.

## Data Availability

The data presented in this study are provided in the main text and in the Appendix A.

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
