# Peer review of "Nectin Cell Adhesion Molecule 4 (NECTIN4) Expression in Cutaneous Squamous Cell Carcinoma: A New Therapeutic Target?"

_biomedicines, 2021, doi:10.3390/biomedicines9040355_

Round 1

Reviewer 1 Report

A very interesting original paper on Nectin 4 expression in cutaneous squamous cell carcinoma. Nectin 4 seems increased in a variety of cancers, as its role seems related to enhanced vascularization and angiogenesis. Quick research in the literature showed that nectin 4 increase was already related to various other squamous cell carcinomas and other cutaneous tumors. However, this is the first study investigating Nectin 4 in cSCC. Only minor queries:

I would probably change the title: "Nectin cell adhesion molecule 4 (NECTIN4) expression in cutaneous squamous cell carcinoma: a new therapeutic target?"

Page 1 line 34; you should add: "Moreover, other environmental exposures namely pollutants, ionizing radiations, and heavy metals are also possible pathogenic factors for SCC" and cite an article such as doi: 10.23736/S0392-0488.20.06600-6. 

Thank You

Reviewer 2 Report

The authors are presenting a basic research breakthrough in terms of establishing a link between expression of NECTIN4 (knockdown by silencing) and cell viability/cell proliferation in cSCC tumor cells, which has not been reported earlier. This fundamental finding lead to suggestion of NECTIN4 as potential target for cSCC treatment. This link has been shown in A341 cells and cSCC cells isolated from biopsies of 34 patients. The manuscript shows good and solid experimental design and set-up, followed by good data presentation, showing correct silencing of the NECTIN4, followed by a decrease in NECTIN4 protein expression, dramatic decrease in A341 cell viability, certain decrease in proliferation, as well as in cell-cell adherence (assessed by the increase in area size and decrease in circularity) and motility decrease.

Influence of NECTIN4 silencing on Akt- and Erk-mediated pathways has been assessed by the same set of methods and the data were consequently discussed as an effect on Erk rather than on Akt.

Statistical evaluation of the data needs a pre-evaluatory assessment of distribution.

Since the silencing of NECTIN4 primarily affects cell viability, to a much higher extent than cell proliferation, this should be included in the discussion, also in the light of therapeuthic potential.

Did the authors test the data's distribution?- If so please state. otherwise reconsider t-test

Many changes used as a base to illustrate the effect of silencing are about 20% except for migration. Proliferation changes ca 15%. Is this sufficient to claim a candidate for a therapy? Maybe it should be commented on in the Discussion as modest but significant changes that can be emphasized in future for purposes of therapy.

For 3 experiments, the statistical significance is remarkably high and the standard deviations are very small -please recheck

Methods

A short rationale for using particular methods would greatly contribute to transparence of otherwise very well postulated methods. For instance, 2.1 and/or 2.4: rationale for the use of MMAE should be provided. Same for ERk inhibition and for the CCK-8 Assay (one sentence would be sufficient). This goes for other parts too.

2.6 Please provide a table for the primer specifics rather than presenting the sequences in the text.

2.7 Please provide a table with antibodies specifics instead of text

2.10 Please shortly state the aim of using patient’s tissue

2.12 Please provide a reference for the criteria used in this evaluation. If the criteria have been determined by the authors, please state this in 2.12

2.13 Statistical Analysis should be numbered as 2.13

Have the normality tests been pre-run (as a pre-requisite for the t-test)? If yes, please provide the corresponding statement. If not, they should be performed to determine normal distribution of the data. In case that the distribution is other than normal, please perform non-parametric tests to determine the significance of the differences.

Results

3.1 Please check the data in the following statement: The mean age of the patients was 81.4 years (range 29–99 years). If thuis is correct maybe median is more appropriate

3.2 line 246-248 This statement speaks rather of a coincidence between NESTIN4 expression and sensitivity to MMAE. It needs statistical assessment. Also, the sentence is dealing with interpretation of the data and should be moved to the Discussion section.

The findings that A431 246 cells express NECTIN4 and are highly sensitive to MMAE imply that these cells may be 247 sensitive to NECTIN4-targeted ADCs such as enfortumab vedotin.

3.2 Line 255, Figure 2. Text: Experiments were 254 performed in five wells for each concentration and repeated three times.

Please state the numer of biological experiments and then the number of technical repetitions, throughout the manuscript. If your initial statement already complies to this, please ignore the comment.

3.3 Please provide differences in percents

3.5 This part is intensively dealing with interpreting the data. It should be divided into the presentation of the data for the purposes of Results Section and for the interpretation, which should be moved to the Discussion.

Discussion

First sentence (line 349) needs to be provided with a reference.

355-366 The changes in adherence have been assessed based on aggregation of spheroids. Are there any available data on adherence changes in 2D culture (cell culture plastic)? It would strengthen the statement and hint towards specific receptors that may be affected by the silencing of NECTIN4.

Line 417 Please re-phrase the following sentence in comparative form: A431 cells may react readily and strongly with 417 antimitotic drugs compared with NHEKs.

The most affected feature in A341 cells upon NECTIN4 knockdown was the cell viability, not the proliferation. An appropriate statement about viability being affected more than the proliferative capacity should be introduced to the Discussion.
